# Integrating across neuroimaging modalities boosts prediction accuracy of cognitive ability

**Javier Rasero**[1]*, **Amy Isabella Sentis**[2,3], **Fang-Cheng Yeh**[3,4‡],
**Timothy Verstynen**[1,4,5‡]*

**1** Department of Psychology, Carnegie Mellon University, Pittsburgh, Pennsylvania, United States of America, **2** Carnegie Mellon Neuroscience Institute, University of Pittsburgh and Carnegie Mellon University, Pittsburgh, Pennsylvania, United States of America, **3** Program in Neural Computation, University of Pittsburgh and Carnegie Mellon University, Pittsburgh, Pennsylvania, United States of America, **4** Department of Neurological Surgery, University of Pittsburgh Medical Center, Pittsburgh, Pennsylvania, United States of America, **5** Biomedical Engineering, Carnegie Mellon University, Pittsburgh, Pennsylvania, United States of America

‡ F-CY and TV are co-senior authors.
* jrasero.daparte@gmail.com (JR); timothyv@andrew.cmu.edu (TV)

**Data Availability Statement:** The data used and generated in this study are available in https://figshare.com/s/b97d2d1ba359e6458cb5. The code used to generate all the results and plots in this study is available in

## Abstract

Variation in cognitive ability arises from subtle differences in underlying neural architecture. Understanding and predicting individual variability in cognition from the differences in brain networks requires harnessing the unique variance captured by different neuroimaging modalities. Here we adopted a multi-level machine learning approach that combines diffusion, functional, and structural MRI data from the Human Connectome Project (N = 1050) to provide unitary prediction models of various cognitive abilities: global cognitive function, fluid intelligence, crystallized intelligence, impulsivity, spatial orientation, verbal episodic memory and sustained attention. Out-of-sample predictions of each cognitive score were first generated using a sparsity-constrained principal component regression on individual neuroimaging modalities. These individual predictions were then aggregated and submitted to a LASSO estimator that removed redundant variability across channels. This stacked prediction led to a significant improvement in accuracy, relative to the best single modality predictions (approximately 1% to more than 3% boost in variance explained), across a majority of the cognitive abilities tested. Further analysis found that diffusion and brain surface properties contribute the most to the predictive power. Our findings establish a lower bound to predict individual differences in cognition using multiple neuroimaging measures of brain architecture, both structural and functional, quantify the relative predictive power of the different imaging modalities, and reveal how each modality provides unique and complementary information about individual differences in cognitive function.

## Author summary

Cognition is a complex and interconnected process whose underlying mechanisms are still unclear. In order to unravel this question, studies usually look at one neuroimaging modality (e.g., functional MRI) and associate the observed brain properties with individual differences in cognitive performance. However, this approach is limiting because it

https://github.com/CoAxLab/multimodal-predict-cognition. The weight maps for the local connectome, cortical surface area and thickness, and sub-cortical volumetric features have been uploaded as NIFTI files to https://identifiers.org/neurovault.collection:9272.

**Funding:** This work was supported, in part, by NIH Award R56MH113634 to FCY.

**Competing interests:** The authors have declared that no competing interests exist.

fails to incorporate other sources of brain information and does not generalize well to new data. Here we tackled both problems by using out-of-sample testing and a multi-level learning approach that can efficiently integrate across simultaneous brain measurements. We tested this scenario by evaluating individual differences across several cognitive domains, using five measures that represent morphological, functional and structural aspects of the brain network architecture. We predicted individual cognitive differences using each brain property group separately and then stacked these predictions, forming a new matrix with as many columns as separate brain measurements, that was then fit using a regularized regression model that isolated unique information among modalities and substantially helped enhance prediction accuracy across most of the cognitive domains. This holistic approach provides a framework for capturing non-redundant variability across different imaging modalities, opening a window to easily incorporate more sources of brain information to further understand cognitive function.

## Introduction

Cognitive abilities are not modularly localized to individual brain areas, but rely on complex operations that are distributed across disparate brain systems (e.g., [1]). Prior work on the association between macroscopic brain systems and individual differences in cognitive ability has, by and large, relied on correlational analyses that usually assess linear changes in a particular cognitive task or measure (e.g., general intelligence quotient) that coincide with specific brain properties such as region size [2, 3], gray matter [4] and white matter [5] volume, cortical thickness [6] and surface area [7], resting-state functional connectivity [8], task-related activity [9], global functional network properties [10], white matter connectivity [11], and other unimodal measures. However, these correlation approaches, based on unimodal imaging methods, suffer several critical limitations. First, due to the mass univariate nature of the analyses, a large number of statistical tests is usually performed, thereby raising the chances of Type I error (false positives) and decreasing the statistical power of the study after adjusting for multiple testing. Second, they do not take into account the mutual dependencies between brain features and therefore ignore redundant sources of variability. Finally, the lack of out-of-sample validation tests leads to over-optimistic results (*i.e*., potential overfitting), thus lowering their generalizability across studies and applicability in clinical routines.

To address some of these limitations, recent studies have adopted machine learning frameworks that can accommodate all of these deficiencies by building predictive models from multivariate features across the whole brain and testing them on independent hold-out data samples. These methodologies have been widely applied to predict cognitive performance (see [12] and references therein) in out-of-sample test sets and have proven particularly popular with resting-state functional connectivity paradigms due to their inherent multivariate nature. For example, recent studies show that functional connectivity profiles, distributed across the brain, can predict up to 20% of the variance in general intelligence [13] and 25% in fluid intelligence, with regions within the frontoparietal network displaying a positive correlation and regions in the default mode network an anti-correlation [14]. Similar sparse regression models have shown how variability in white matter integrity of association pathways can reliably predict individual differences in cognitive ability [15]. By building predictive models that can be evaluated in out-of-sample test sets, as opposed to simple association analyses, these machine learning approaches can quantify the degree of generalizability of particular findings, providing key insights into potential for neuroimaging based biomarkers for cognitive function.

Nevertheless, these multivariate methods do suffer from problems with interpretability. For example, in the pursuit of maximizing performance, some approaches may rely on complex non-linear models (e.g., deep neural networks), for which directly assessing feature importance can be challenging. Even in relatively simple multivariate linear models, which establish a transparent relation between the input features and the response variables under investigation (e.g., sensory, cognitive or task conditions), large weights that carry no signal-of-interest whatsoever can emerge [16]. In this regard, univariate methods are more straightforward to interpret and therefore, it has been argued that multivariate and univariate analyses should be considered complementary when exploring brain-behavior associations [17].

Despite the success in applying predictive modeling approaches to the mapping of brain systems to individual differences in cognitive performance, previous work has largely focused on unimodal methods, which may not be sufficient to capture enough neurobiological properties due to the fact that different neuroimaging modalities reveal fundamentally distinct properties of underlying neural tissue. For example, functional MRI (fMRI) and diffusion MRI (dMRI) reveal separate, but complementary, properties of the underlying connectome that independently associate with different aspects of cognition [18]. This means that different measures of brain structure and function may reveal complementary associations with cognitive abilities that collectively boost the power of predictive models.

One of the challenges of building multimodal models of individual differences is the increased complexity of the explanatory model when one attempts to combine all the sources of variation. Modeling variability from a single neuroimaging modality is an already high dimensional statistical problem [19–21], with many more features than observations. Adding more modalities exponentially increases model complexity, increasing the risk of overfitting, even when traditional approaches to dimensionality reduction (e.g., principal component regression) or sparse feature selection (e.g., LASSO regression) are applied. One way around this dimensionality problem is transmodal learning [22], a multi-modal predictive approach that combines elements from transfer [23] and stacking (sometimes also called stacked generalization) [24] learning paradigms. Transmodal learning takes independent predictions from separate channels (e.g., generated from separate imaging modalities) and runs a second model using the single-channel predictions as the inputs. This second "stacked" model attempts to find unique sources of variability in the different input channels. Redundancy in variance, *i.e.*, if two different imaging modalities are picking up on the same sources of variability, is accounted for through the use of feature selection methods. The end result is a more holistic prediction model that tries to explain more variance than individual input channels. Such a transmodal learning approach was recently shown to be effective at integrating structural and functional MRI measures to generate a reliable prediction of participant age whose residuals also explained individual differences in objective cognitive impairment [25].

In the present study, we used the transmodal, or stacked, learning method to quantify the extent to which the combination of data from multiple neuroimaging modalities permits increasing predictive performance in several cognitive domains, including intelligence, sustained attention, working memory, spatial orientation and impulsivity. By using a large dataset, comprised of multiple neuroimaging measures from 1050 subjects from the Human Connectome Project [26], we demonstrate that for most cognitive domains a significant enhancement in overall prediction capacity is achieved when multiple modalities are integrated together, indicating that each brain measurement provides unique information about the underlying neural substrates relevant for cognitive function. In addition, this analysis yields a multi-modal weight pattern for each cognitive ability, namely, the subset of architectural brain features whose combination yields a significant and complementary enhancement of the prediction performance.

## Results

Our primary goal was to see if predictions that integrate across neuroimaging modalities provide a boost to the prediction capability of individual differences in cognitive ability. For purposes of our analysis, the primary neural measures consisted of MRI-based assessments of (1) functional networks defined as Fisher's z-transformed Pearson correlation coefficients between resting BOLD time series, (2) measures of cortical surface area, (3) cortical thickness attributes, (4) global and subcortical volumetric information and (5) local connectome features representing the voxel-wise pattern of water diffusion in white matter. Our multi-level, stacked modeling approach (see Fig 1 and Material and methods section for details) uses a L1-constrained (LASSO) variant of principal component regression (PCR) to generate predictions of specific cognitive scores from single imaging modalities in a training set. These are referred to as *single-channel* models. To integrate across modalities, we stacked these single-channel predictions together and used them as inputs to a separate LASSO regression model that performs a weighted feature selection across channels. This is referred to as the *stacked* model and it produces a new set of predictions for cognitive scores of individuals by selecting and reweighting the individual channel predictions. The use of a LASSO model at this new learning level even with only five features (the number of single-channels) guaranteed that redundant modalities did not contribute to the final predictions. Performance of the single-channel and stacked models are then evaluated by comparing the observed scores with the predicted scores in the out-of-sample sets. All models were fit on 70% of the data (training set) and tested on the remaining 30% (test set). A Monte Carlo cross-validation procedure [27] with 100 random stratified splits was employed to assess the generalization of these predictions.

The overall performance of the single-channel and stacked models are depicted in Fig 2. These accuracies were determined using the coefficient of determination $R^2$ (see S1 Fig for the mean absolute error scores), which shows the percent variance explained by each model in out-of-sample test sets. In Fig 3, the contributions of each channel to the stacked model (estimated by the LASSO weights) are displayed for those domains in which stacking bonus is positive at the 95% confidence level, *i.e.* the scenario in which different measurements aggregate complementary and non-redundant variability. Finally, in order to understand the relative feature importance in the predictions of each brain measurement, we refitted the LASSO-PCR estimator to each single-channel model using all observations. This allowed us to access the pattern of feature weights estimated within each individual channel. As detailed in the Material and Methods section and following the recommendations given in [16], these were each further multiplied by their input data covariance matrix, to approximate the feature contributions as encoding weights. The resulting maps are depicted in Fig 4, only for those measurements whose median cross-validated contribution to the stacked model is different from zero at $\alpha = 0.05$ significance level. In the following sub-sections we shall elaborate on the specific pattern of results for each cognitive factor represented by the scores given in Table 1.

### Global cognitive function

Global cognitive function was estimated by the Composite Cognitive Function score, a proxy for a general estimate of intelligence. Here the single-channel models based on cortical surface area and local connectome features produced the highest predictive rates for individual modalities, with a median $R^2 = 0.049$, 95% CI [0.040, 0.055] and 0.049, 95% CI [0.043, 0.052] respectively. Moreover, the relative prediction accuracy of these two models did not differ statistically (one tailed Wilcoxon test $p = 0.429$, rank-biserial correlation $\frac{W}{S} = 0.021$). Compared

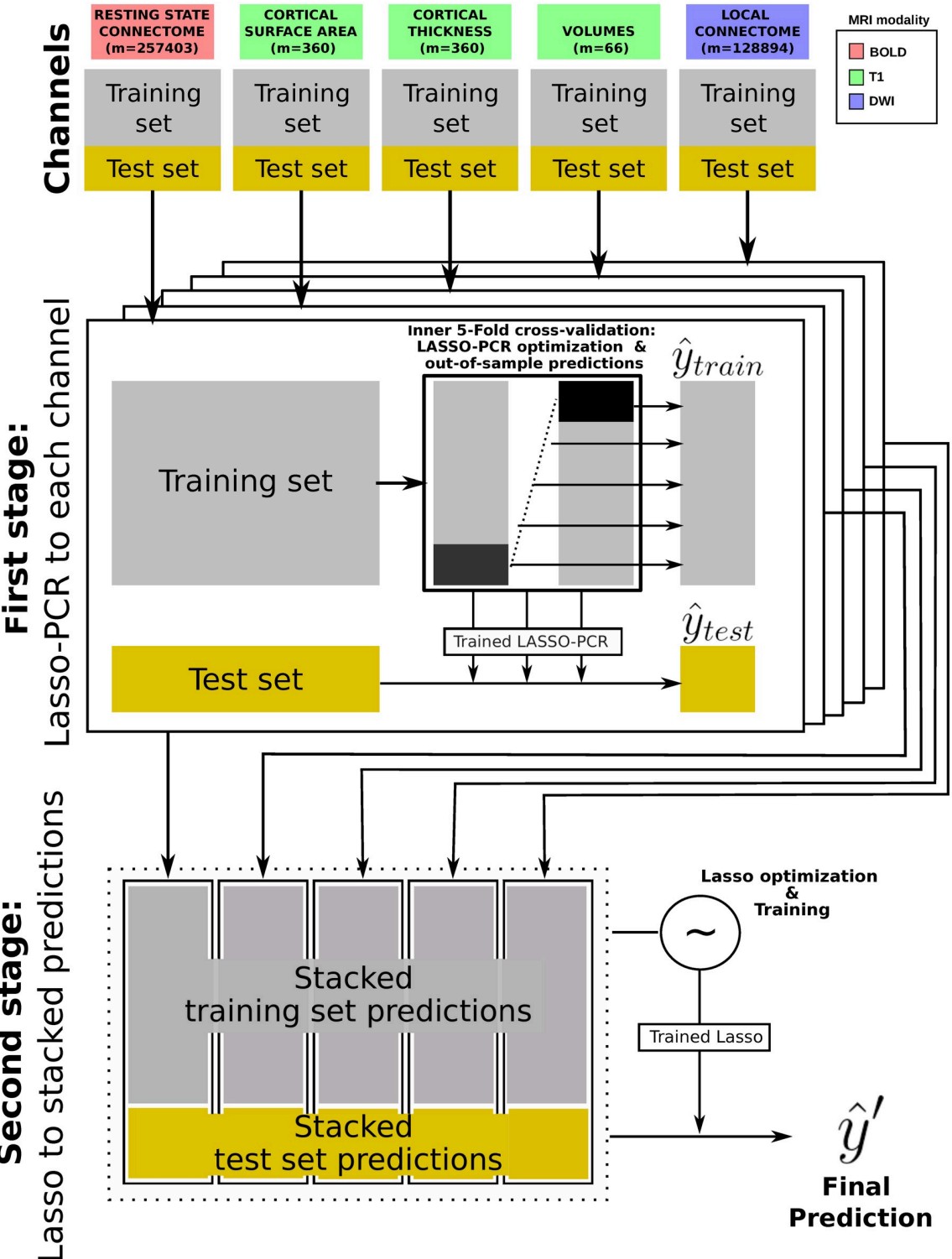

**Fig 1. Stacking methodology for multi-modal data prediction.** In the first step and for each brain measurement, a 5-fold cross-validation is applied to the training set to simultaneously optimize a LASSO-PCR model and produce out-of-sample training set predictions. The optimized trained LASSO-PCR model is then used to generate predictions from the test set. In the second learning step, training and test set predictions are stacked across channels. A new LASSO model acting on the new training set matrix is then optimized with an inner 5-fold cross-validation and fitted to generate the final predictions on the new test set.

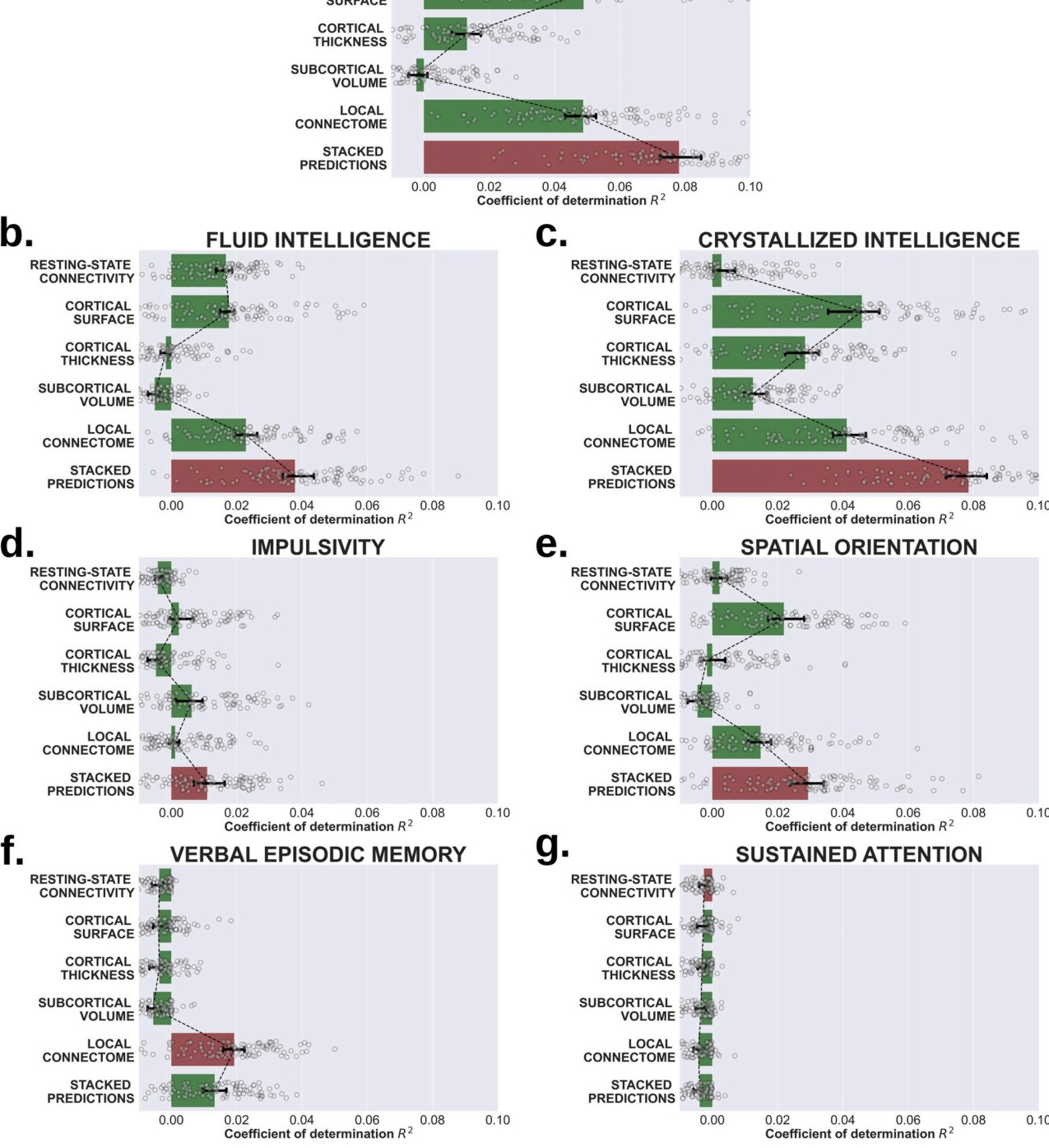

**Fig 2. Single-channel and stacked performances to predict cognition.** The coefficient of determination, $R^2$, between the observed and predicted values of seven cognitive scores using each brain measurement separately and together by stacking their predictions. The scenario that yields the maximum predictive accuracy in out-of-sample tests is shown in red.

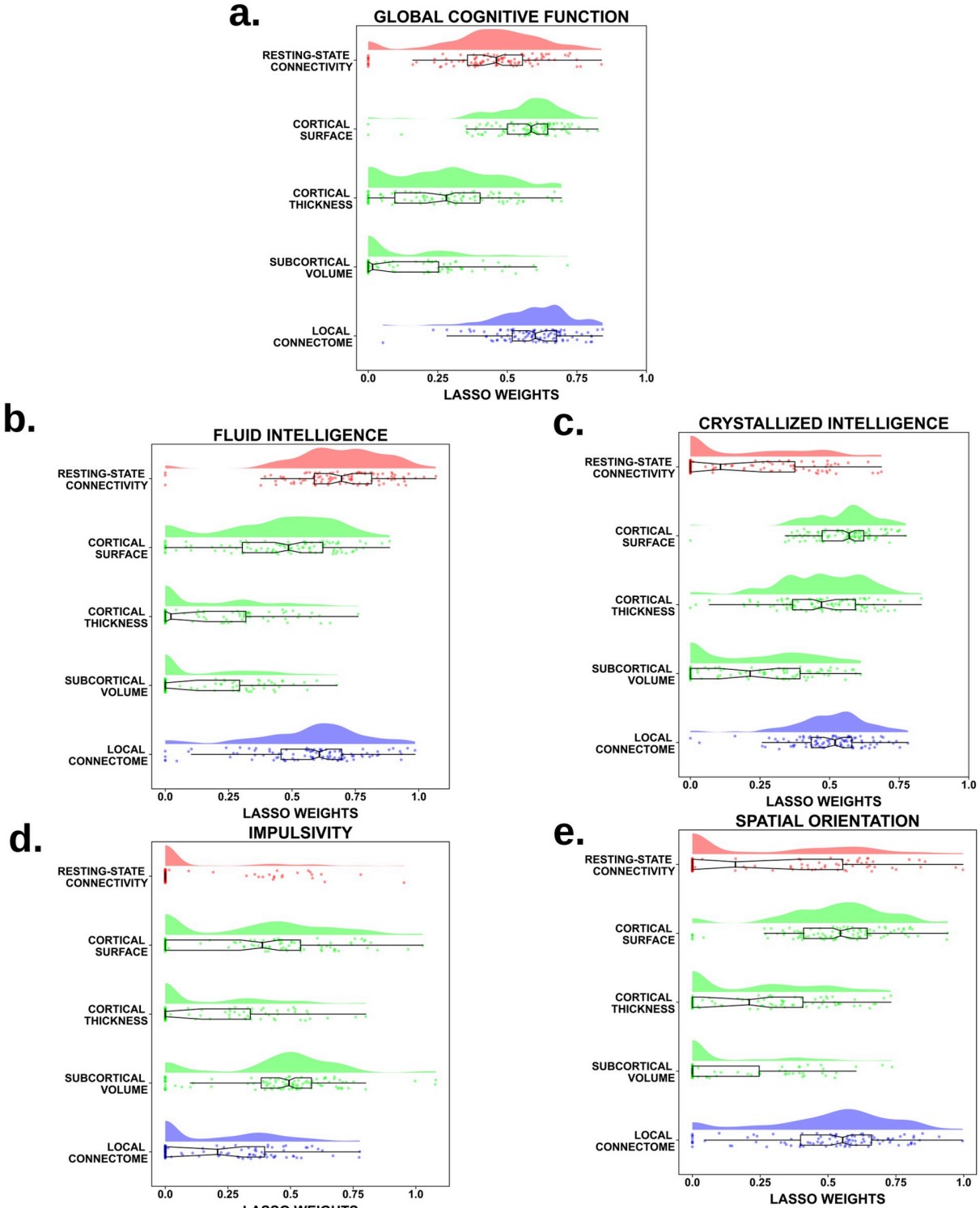

**Fig 3. Regression coefficient distribution of each single-channel in the stacked model.** Across the 100 different data splits, the weight distribution assigned to the out-of-sample predictions of each brain measurement by the stacked LASSO model in those cognitive scores in which stacking significantly improved the overall performance.

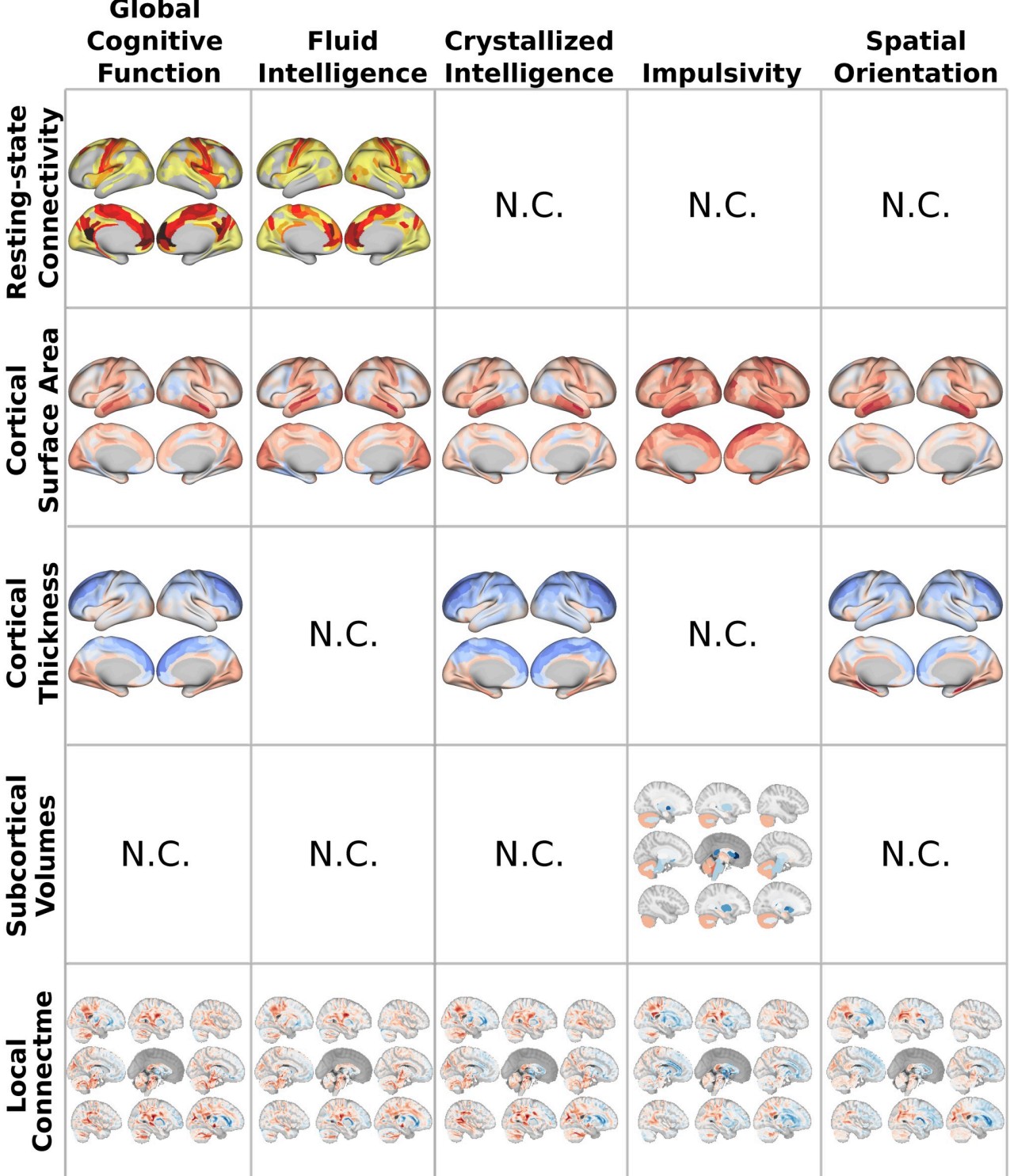

**Fig 4. Multi-modal neuroimaging patterns from cognitive prediction.** Encoding weight maps of each brain measurement whose contribution to predicting cognitive scores during stacking is statistically non-redundant (N.C. ≡ Non-Contributing). Red and blue colors in the brain images (*i.e.*, local connectome, cortical surface area, cortical thickness, and subcortical volumes) display positive and negative weights respectively. For the resting-state connectivity features the strength maps are instead displayed, estimated as the sum over the rows (or columns) in the absolute matrix of links' weights, thresholded to concentrate only on their 1% largest values. The weight map for the volumetric properties in Crystallized Intelligence is marked as non-contributing (N.C.) since this channel did not survive after adding the confounders channel to the stacked model.

**Table 1. Main descriptive statistics for the response cognitive variables.**

| Name of the test: measure score | Mean | Median | Skewness | Kurtosis | Mild (Extreme) % outliers | Lower-Higher 95% mean CI |
|---|---|---|---|---|---|---|
| NIH Toolbox Cognition: Total Composite Score | 122.28 | 120.77 | 0.21 | -0.49 | 0.00 (0.00) | 121.37-123.17 |
| NIH Toolbox Cognition: Fluid Composite score | 115.43 | 114.15 | 0.26 | -0.46 | 0.00 (0.00) | 114.72–116.13 |
| NIH toolbox Cognition: Crystallized Composite score | 117.90 | 117.81 | 0.11 | 0.09 | 0.01 (0.00) | 117.29–118.50 |
| Delay Discounting Test: AUC for discounting of $200 | 0.26 | 0.20 | 1.33 | 1.57 | 0.04 (0.00) | 0.25–0.27 |
| Variable Short Penn Line Orientation Test: total number of correct responses | 14.93 | 15.00 | -0.27 | -0.14 | 0.00 (0.00) | 14.67–15.23 |
| Penn Word Memory Test: total number of correct responses | 35.64 | 36.00 | -0.82 | 0.57 | 0.01 (0.00) | 35.45–35.81 |
| Short Penn Continuous Performance Test: sensitivity | 0.96 | 0.97 | -3.29 | 19.76 | 0.04 (0.01) | 0.95–0.96 |

to the cortical surface area and local connectome models, a significant drop in prediction performance occurred for resting-state connectivity (median $R^2$ = 0.016, 95% CI [0.014, 0.020]), cortical thickness (median $R^2$ = 0.013, 95% CI [0.009, 0.017]) and global and sub-cortical volumetric (median $R^2$ = −0.002, 95% CI [-0.005, 0.001]) features. Thus we see substantial variability across individual neuroimaging modalities in their predictive utility on a measure of global cognitive function.

After integrating predictions across modalities, however, an important improvement in overall accuracy is observed. The stacked model raised the median $R^2$ to approximately 0.078, 95% CI [0.072, 0.084] in global cognitive function. Thus, the stacked model predicted a significant incremental median bonus $\mathcal{B}$ = 0.030, 95% CI [0.025, 0.035] compared to the best single-channel model.

Based on the LASSO weights in the stacked model (see Fig 3A), we identified the local connectome and cortical surface areas as the strongest contributing measurements, with the former (median $\beta$ = 0.601, 95% CI [0.582, 0.640]) contributing more than the latter (median $\beta$ = 0.586, 95% CI [0.572, 0.608]). Interestingly, resting-state connectivity was still a reliable predictor (median $\beta$ = 0.461, 95% CI [0.412, 0.483]), as was cortical thickness (median $\beta$ = 0.281, 95% CI [0.203, 0.312]), although to a lesser degree. A median weight not statistically different from zero (at the 0.05 significance level) assigned to the volume channel predictions showed that these factors did not appear to reliably contribute to the stacked model.

For the largest contributing channel, the local connectome, global cognitive ability particularly associated in a positive way (warmer colors in Fig 4, first column) with signal in cranial nerves and fibers along the rubrospinal and the central tegmental tracts. In contrast, fibre bundles in the brainstems like the medial lemniscus and medial longitudinal fasciculus and commissural pathways like the anterior commisure were negatively associated with global cognitive ability (cooler colors). Estimated average positive and negative loadings for an extensive list of white matter tracts in a population-based atlas of the sctructural connectome [28] can be found in S1 and S2 Tables. For cortical surface area, regions in the ventral temporal lobe, the somatosensory and the visual cortex were largely positively associated with global cognition while negative associations were more scarce, mainly concentrating on areas of the posterior multimodal network. Interestingly, a different pattern was observed for cortical thickness, with a more pronounced emergence of negative associations, that extend over the prefrontal cortex. Finally, resting-state connectivity to the visual cortex and within the frontoparietal network appeared to be positive associated with global cognition, whereas the largest contributions are negative associations from links connecting to the default-mode network (see also S2 Fig). Loadings for the volumetric features are not shown because this channel's predictions did not survive the feature selection step in the stacked LASSO model.

## Fluid intelligence

Global cognitive ability is usually decomposed into two domains [29]: fluid intelligence (*i.e.*, the ability to flexibly reason on new information) and crystallized intelligence (*i.e.*, the ability to utilize the knowledge and skills acquired through prior learning experience). Thus we next wanted to determine how similar or different the prediction models were for these two sub-components of general cognitive ability are. For fluid intelligence, extracted from the NIH toolbox Cognitive Fluid Composite Score, local connectome fingerprints yielded the highest coefficients of determination (median $R^2 = 0.023$, 95% CI [0.020, 0.026]), significantly exceeding those from cortical surface areas (median $R^2 = 0.018$, 95% CI [0.013, 0.019]) and resting-state connectivity (median $R^2 = 0.017$, 95% CI [0.013, 0.019]). Both cortical thickness and volumetric features failed to predict statistically fluid intelligence (negative to null $R^2$ at the 95% confidence level). Stacked predictions raised the variability explained to a median $R^2 = 0.038$, 95% CI [0.034, 0.043], which is translated into a 0.018, 95% CI [0.014, 0.020] of expected median stacking bonus $\mathcal{B}$.

Interestingly, despite not showing the largest predictive accuracy of fluid intelligence at the single-channel level, resting-state connectivity features supplied the largest portion of non-redundant variability to the stacked model (median $\beta = 0.695$, 95%CI [0.649, 0.739]), followed by the local connectome ($\beta = 0.608$, 95% CI [0.576, 0.644]) and cortical surface area factors ($\beta = 0.486$, 95% CI [0.437, 0.541]). On the other hand, cortical thickness attributes and volumetric information appeared to provide a null contribution to the stacked model (see Fig 3B).

Since fluid intelligence is one of the two components of the global cognitive ability score (Pearson correlation coefficient with the global cognitive function score $r = 0.849$), it is not surprising that the weight maps in both domains showed large correlations across the five modalities (see S3 Fig). Regardless of this, subtle phenotypic differences can still be observed (see Fig 4, second column); namely, the predominance of large positive loadings in the brainstem nerves over the cranial nerves (see S1 and S2 Tables), the predominance of resting-state links connecting to the medial prefrontal cortex and the emergence of more positive correlations to the visual and somatomotor cortex (see also S2 Fig), and the relative increase of negative weighs for cortical surface areas in the inferior temporal and fusiform gyri. Loadings for global and subcortical volumes and cortical thickness features are not shown because they did not survive the feature selection step in the stacked LASSO model.

## Crystallized intelligence

For crystallized intelligence we extracted the NIH toolbox Cognition Crystallized Composite Score. In this domain, the highest predictive accuracies were provided by cortical surface attributes (median $R^2 = 0.046$, 95% CI [0.036, 0.051]), statistically comparable (one tailed Wilcoxon test $p = 0.512$, rank-biserial correlation $\frac{W}{S} = 0.003$) with the performance from local connectome features (median $R^2 = 0.041$, 95% CI [0.037, 0.047]). These were followed by cortical thickness (median $R^2 = 0.028$, 95% CI [0.022, 0.033]), volumetric measurements (median $R^2 = 0.012$, 95% CI [0.09, 0.017]) and resting-state connectivity (median $R^2 = 0.003$, 95% CI [0.0, 0.007]). By stacking these predictions, we obtained a median bonus $\mathcal{B} = 0.034$, 95% CI [0.029, 0.038], with the stacked model reaching a median $R^2 = 0.078$, 95% CI [0.072, 0.083].

In contrast to global cognition, variability in the stacked model was foremostly driven by cortical surface area factors (median $\beta = 0.571$, 95% CI [0.548, 0.590]), which significantly exceeded the contribution from local connectome (median $\beta = 0.520$, 95% CI [0.487, 0.547]), cortical thickness (median $\beta = 0.471$, 95% CI [0.445, 0.522]) and volumetric properties (median $\beta = 0.215$, 95% CI [0.105, 0.316]). Likewise, the stacked LASSO model shrunk away

predictions from resting-state connectivity attributes at the 95% confidence level when combined with the rest of channels (see Fig 3C).

Similarly to the fluid intelligence domain, multi-modal weight patterns of crystallized intelligence (see Fig 4, third column) resembled those of global cognition (Pearson correlation between crystallized intelligence score and global cognitive function score $r = 0.769$), with main positive associations along fibre bundles in the brainstem which include the medial longitudinal fasciculus and central tegmental tracts and cranial nerves. Interestingly, for this domain there is an enhanced negative association with some association pathways represented by the bilateral cingulum tract (see also S1 and S2 Tables). In addition, a predominance of the middle and inferior temporal gyri were found for cortical surface areas and an increase in negative associations with cortical thickness in regions of the frontal cortex.

## Impulsivity

One factor not reliably measured in the Composite Cognitive Function score is impulsivity or self-regulation, *i.e.*, the ability to suppress contextually inappropriate behaviors. To assess this we extracted the area-under-the-curve (AUC) for discounting of $200 from the Delayed Discounting Task. Even though the percent variance explained by the single-channel models for this impulsivity measure were low (see Fig 2D), their performance improved using the stacked predictions (median $R^2 = 0.011$, 95% CI [0.008, 0.016]). Indeed, the stacked model performance exceeded that of the best single-channel, in this case volumetric factors (median $R^2 = 0.006$, 95% CI [0.001, 0.010]), translated into a stacking bonus $\mathcal{B} = 0.007$, 95% CI [0.002, 0.010]. Such stacking improvement took place even though the rest of the channels individually failed to predict better than simply the average impulsivity score ($R^2 = 0$) at a 95% confidence level.

Interestingly, unlike the models for global cognition and intelligence, for predicting impulsivity not only did the volumetric factors survive the LASSO feature selection step but they emerged as the most important explanatory source of variability in the final out-of-sample predictions (median $\beta = 0.494$, 95% CI [0.459, 0.518]), which exceed those from cortical surface areas (median $\beta = 0.388$, 95% CI [0.308, 0.454]) and local connectome properties (median $\beta = 0.209$, 95% CI [0.005, 0.309]). Resting-state connectivity and cortical thickness attributes appeared to play a negligible role at the 95% confidence level in combination with the aforementioned measurements in the stacked model (see Fig 3D).

Impulsivity weight maps for these contributing channels are depicted in Fig 4, fourth column. Subcortical attributes showed a strong positive influence of cerebellar cortex and the amygdala and negative loadings in the brainstem, putamen and left nucleus accumbens. The importance of the remaining volumetric features can be found in S3 Table, revealing an overall positive association of cortical volumetric measures with impulsivity scores. With respect to cortical surface areas, loadings were mainly positive, with greatest associations found in areas of the medial frontal gyrus, the middle and inferior temporal gyri and the somatosensory cortex. Finally, particularly important areas of local connectome features positively correlated with impulsivity were found along fibers from the brainstem that include the dorsal and longitudinal fasciculus, projection pathways like the occipitopontine tracts and cerebral pathways represented by the superior cerebellar peduncle (see also S1 Table). In contrast, the largest negative loadings were located along the fornix, cingulum and medial lemniscus tracts (see also S2 Table).

## Spatial orientation

Another cognitive domain not covered under global cognitive functioning is spatial orientation, which is the ability to reference how the body or other objects are oriented in the

environment and reflects a critical cognitive domain for spatial awareness. We extracted scores from the Variable Short Penn Line Orientation Test to look at individual differences in spatial orientation ability. The predictive models for spatial orientation followed the similar tendency observed in our other models thus far, with the stacked predictions improving overall single-channel accuracies. The best performing single-channel model was for cortical surface area (median $R^2$ = 0.022, 95% CI [0.017, 0.028]), which was greater than the local connectome (median $R^2$ = 0.015, 95% CI [0.011, 0.018]). The rest of measurements failed to predict better than $R^2$ = 0 at the 95% confidence level.

Like the previous cognitive measures, the prediction of individual differences in spatial orientation improved when modalities were integrated together, with the performance of the stacked model being $R^2$ = 0.029, 95% CI [0.024, 0.034] and giving rise to a stacking bonus $\mathcal{B}$ = 0.009, 95% CI [0.005, 0.012] with respect to the best single-channel model. As shown in Fig 3E, the stacked model eliminated the contributions of resting-state connectivity and volumetric features, suggesting that these factors did not provide unique contributions to predicting spatial orientation ability above that of the cortical surface (median $\beta$ = 0.546, 95% CI [0.515, 0.589]) and white matter measures (median $\beta$ = 0.554, 95% CI [0.508, 0.584]). Interestingly, even though its predictive power as a single channel was poor, cortical thickness features appeared to provide a small but non-redundant contribution to the stacked model (median $\beta$ = 0.209, 95% CI [0.023, 0.289]).

Finally, weight maps estimated from these contributing channels are displayed in Fig 4, fifth column. Interestingly, we can appreciate a positive correlation with local connectome features particularly along some projection pathways connecting to regions in the occipital cortex (optical radiation, central tegmental and occipito-pontine tracts), which are involved in visual demanding tasks, whereas negative associations are mainly dominated by the frontopontine tract and fibres bundles in the association pathways including the frontal aslant tract and cingulum. Regarding structural cortical attributes, positive loadings from surface area factors were found in regions along the inferior and middle temporal gyri and in the somatomotor network, whereas negative associations took place in parts of the precuneus like the parieto-occipital fissure. On the other hand, except for regions in the visual cortex and the parahippocampal gyrus, thickness properties exhibited a negative correlation with spatial orientation that spans over the entire brain cortex.

## Verbal episodic memory

The Penn Word Memory test captures verbal memory abilities. For this response variable, stacked prediction accuracy (median $R^2$ = 0.013, 95% CI [0.010, 0.017]) did not improve the single-channel's best performance represented by the local connectome fingerprints (median $R^2$ = 0.019, 95% CI [0.016, 0.023]). The rest of the measurements all exhibited a negative median $R^2$, meaning that they perform worse than using predictions from the mean response variable (see Fig 2F). As a consequence, neither the regression coefficients showing the single-channel contributions to the stacked model nor the weight maps were reported for this cognitive score.

## Sustained attention

The Short Penn Continuous Performance Test is the measure of sustained attention. As shown in Fig 2G, accuracies of single-channels and stacked model in this domain were overall negligible and worse or not statistically different from those provided by the mean response variable. Owing to this, neither the regression coefficients showing the single-channel contributions to the stacked model nor the weight maps were reported for this cognitive score.

### Prediction adjustment for non-neuroimaging confounders

As a post-hoc analysis, in order to test the performance and contribution of each single-channel of brain measurements to the predictions in the presence of other non-neuroimaging confounders, we incorporated age, sex and education level variables together as an additional channel to the stacking procedure.

As expected, compared to the neuroimaging measurements, predictions from this additional channel explained a greater portion of the data across all cognitive domains: crystallized intelligence (median $R^2$ = 0.246, 95% CI [0.228, 0.251]), global cognitive function (median $R^2$ = 0.165, 95% CI [0.156, 0.173]), fluid intelligence (median $R^2$ = 0.065, 95% CI [0.058, 0.069]), impulsivity ($R^2$ = 0.024, 95% CI [0.018, 0.028]) and spatial orientation (median $R^2$ = 0.025, 95% CI [0.020, 0.029]).

Importantly, except for the volumetric factors in crystallized intelligence, the contributions from the neuroimaging channels survived after stacking the predictions with the confounders and indeed, they followed the same pattern in terms of relative importance of the brain measurements found before (see S4 Fig). As a consequence, this persistence of contributing channels helped raise the total variability explained by the neuroimaging and confounder features together: crystallized intelligence (median $R^2$ = 0.270, 95% CI [0.260, 0.279]), global cognitive score (median $R^2$ = 0.197, 95% CI [0.190, 0.210]), fluid intelligence (median $R^2$ = 0.089, 95% CI [0.081, 0.095]), impulsivity ($R^2$ = 0.032, 95% CI [0.028, 0.035]) and spatial orientation (median $R^2$ = 0.045, 95% CI [0.039, 0.050]).

### Testing for ceiling and floor effects

In order to rule out the possibility that ceiling and floor effects of some of the scores might be influencing the poor performances obtained, we repeated the analyses for sustained attention and impulsivity using respectively the Short Penn CPT Median Response Time for True Positive Responses score and the NEO-Five Factor Model (NEO-FFI) score for extraversion, which both appeared to exhibit more friendly distributions (see S5 Fig). Interestingly, for both scores all single-source channels as well as stacked predictions performed worse than random guess ($R^2 < 0$), supporting the effect sizes observed in these domains for the original scores.

## Discussion

Here we tested whether multiple functional, diffusion, and morphological MRI-based measurements of brain architecture constitute complementary sources of information for predicting individual differences in cognitive ability. We accomplished this by means of a stacking approach for multimodal data, a two-level learning framework in which groups of features are first separately trained and their predicted response values subsequently stacked to learn a new model that takes into account redundant effects across channels. Our results show that for most of the cognitive measures tested integrating across different brain measurements provides a boost to prediction accuracy, highlighting how different imaging modalities provide unique information relevant to predicting differences in human cognitive ability.

One of the strengths of our approach is the assessment of how performance in different cognitive domains associates with a wide range of brain measurements. Overall, our results show that effect sizes tend to be moderate (at most explaining less than 10% of the variance after stacking), which nevertheless go in line with a recent research in a large sample size (N = 10,145) reporting a similar degree of variability using cortical morphology information for predicting fluid and crystallized intelligence [30]. Likewise, we adopted a rigorous strategy that takes in consideration the effects of factors that are known to confound predictions of individual differences in behavior and cognition (e.g., brain size and education level). While

conservative, this approach highlights the importance of identifying optimal deconfounding approaches to disentangle brain-behavior associations from spurious effects. Therefore, all this raises the question of where the remaining variability may come from. It might be possible that the metrics employed here, which largely reflect static architectural aspects of global brain systems, are missing the fundamental dynamics of neural circuits during relevant behavioral states for expressing specific cognitive functions. In this regard, specific task-evoked fMRI measurements that directly assess the neural reactivity during cognitive evaluation [31–33] could help raise the overall predictive power. For example, we recently showed that brain activation patterns during affective information processing tasks predict an important portion of individual differences of cardiovascular disease risk factors, a finding that we could not have reached had not we used the appropriate and specific task fMRI experiment [34]. Additionally, increasing both spatial resolution to better capture features of structural-functional variation [35] and temporal resolution for a more accurate decoding of the underlying brain dynamics [36] could be valuable and complementary sources of cognitive performance correlation. Thus we consider the work here a proof-of-principle for making holistic models that predict specific cognitive abilities, which could be further improved with additional, more specific, inputs.

Of particular note is that the observed effect sizes from resting-state connectivity are consistently small, which appears to be in conflict with previous results that reported a medium-large correlation ($r = 0.5$) between patterns of resting-state connectivity and fluid intelligence functioning [14]. In our case and for this particular domain, the maximum performance that we achieved across all simulations is ostensibly smaller ($R^2 = 0.056$, r = 0.25). Nevertheless, such a decrease in the effect sizes was expected due to our use of a much larger sample size ($N = 1050$ versus $N = 126$), which reduces inflated results caused by sampling variability and therefore, findings are more reproducible and inferred patterns generalizable to a broader population spectrum [37]. In addition, it is important to note that our preprocessing pipeline does not include a global signal regression step, which is supposed to improve resting-state functional connectivity based behavioral prediction accuracies [38]. This step is still controversial since it is not clear whether it supplies real or spurious information [39]. Finally, we relied on the coefficient of determination, $R^2$, to assess the predictive power of the learned models. For regression tasks, this performance metric is recommended over the usual Pearson correlation coefficient, which overestimates the association between predicted and observed values [40, 41].

The overall stacking approach to multimodal integration that we applied here closely follows work by Liem and colleagues [25], who used a stacking approach with multimodal brain imaging data to improve the performance in individual age prediction, although with two big differences in structure. First is the choice of the algorithm for the second learning stage in this transmodal approach. Albeit performances from a random forest algorithm, as used by Liem and colleagues, would have been proven to be less variable compared to other well known algorithms in similar scenarios [42], we decided to use a LASSO regression model because of its simplicity (it only has one hyperparameter to tune) and due to the fact that the L1 penalty term can automatically get rid of the redundant variability of the different channels. The second difference is the number of neuroimaging modalities, since we have also included diffusion data in our study. Indeed, we have demonstrated that the inclusion of local connectome features played an important role for prediction, since they alone account for a moderate rate of variability consistent across all cognitive domains. Moreover, such variability survives and indeed prevails in some domains when combined with the rest of single-channel predictions. This finding validates the role of the local connectome fingerprint as a reliable correlate of cognitive factors at the individual level [15] and suggests its complementary role in combination with other brain measurements. In particular, white matter diffusion tracts provide a putative

structural basis for the macroscopic human connectome that is reflected in the correlation for age [43] and cognition [44, 45]. Furthermore, it is important to stress that local connectome fingerprints do not rely on fiber tracking algorithms, which reduces the risk of false-positive bias when mapping white matter pathways [46–48]. Our findings here with respect to predicting cognitive ability, along with the work of Liem and colleagues on age predictions [25], simply demonstrate how powerful a stacking approach can be to maximizing explainable variance from multiple imaging modalities.

Interestingly, although stacked predictions clearly increase the variability explained at a global cognitive level, this is not the case across all domains. For example, global, fluid, and crystallized intelligence, impulsivity and spatial orientation all show improvements in prediction accuracy, in contrast to attention and word memory function. These results might be caused by the existence of a hierarchical cognitive categorization, with high complex functions demanding the integration of multi-modal aspects of the brain compared to lower level functions [49]. Alternatively, individual differences in some cognitive areas might be mostly parametrized by one specific brain sub-system that captures all the variability. For instance, white matter structure is an important substrate of cognitive performance whose deterioration, notably in the hippocampus, is the first sign of memory decline at both early and late stages in Azheimer's disease [50, 51]. On the other hand, it might also happen that for certain cognitive domains, the measurements considered in this study do not constitute a sizable source of variability and therefore stacking is only aggregating noise to the predictive model. Finally, associations with cognitive performance might be affected by the inherent nature of the cognitive tests, either due to an imperfect design that adds unwanted variability, or because of some properties (e.g., ceiling and floor effects) of the sampling distribution that can make them not suitable for predicting interindividual differences. For example, in our dataset, scores in the Short Penn test display a heavy deviation from normality, that can affect sensitivity in regression models. However, as we also showed, even after replacing these scores for others with more desirable distribution properties, prediction accuracies did not improve, which highlights how challenging it can be to predict individual differences of cognitive performance using neuroimaging data.

On the other hand, it is worthwhile emphasizing the justification of our stacking learning approach compared to, for example, a simple regression model that includes all of the features across all modalities. First, our framework effectively estimates the unique variance explained by each neuroimaging modality in predicting cognitive performance, acting thus as a form of feature selection at the neuroimaging modality level that accounts for redundancies in correlated signals. Second, the distinct underlying noise structure across modalities is handled in our approach by fitting each single-channel independently, in contrast to linear regression models using all the data together. Finally, our predictive framework shares some characteristics with resample-based ensemble methods (e.g., random forest), which in some cases might be a more efficient way of handling wide datasets (number of features exceeding the number of observations). For example, in a supplementary analysis predicting global cognitive function (not shown in Results), we found that the same LASSO-PCR procedure applied to the concatenated data across modalities (a matrix with 387082 features) performed significantly worse (median $R^2 = 0.047$ 95% CI [0.044, 0.052]) than our stacking learning model.

A possible limitation of our study arises from the fact that predefined test scores were employed as response variables, which in most cases are largely coarse measures of cognitive ability that may rely on redundant underlying subprocesses, leading to a degree of similarity in the brain architecture features that contribute to predicting individual differences. Regardless of this, spatial correlation analysis shows that a portion of each cognitive measure's weight map is unique (including for the global cognitive score that is constructed from both

crystallized and fluid intelligence scores) and therefore there exist brain structures whose roles are specific to the area of cognition involved (see S3 Fig). Future studies might benefit from adopting more sensitive approaches to measuring specific cognitive factors (e.g., psychophysical measurements) that can carefully isolate primary cognitive abilities.

Finally, our current framework treats brain measurements separately at a first step, based on the assumption that they represent independent and non-overlapping sources of cognitive variation. As we showed, this is far from true: there exists some degree of redundancy across the imaging modalities. Future studies might attempt to find a decomposition into multidimensional representations of unique and shared variance across brain measurements. This would likely increase the number of single-channels whose individual predictions can be later exploited by our stacking approach. Alternatively, we speculate that a similar scenario as the one employed here could be accommodated using architectures of multi-layered neural networks. These models, however, require very large training sets, with tens of thousands of observations or more, which dwarfs most of the largest neuroimaging data sets currently available. Though, neural network approaches seem to be an appropriate future extension to our study as a stack of modality-wise neural networks, connected to a last hidden layer allowing for inter-channels connections, once appropriately large data sets become available.

Despite these limitations, our work here builds on the growing body of work attempting to integrate information from different neural sources so as to maximize explained variability of individual differences. Our approach predicts individual differences in cognition by separately fitting measurements of structural, functional and diffusion modalities and subsequently stacking predictions to enhance overall accuracy while removing redundant contributions. Even though a large portion of variance in the data remains unaccounted for, our results demonstrate that effect sizes can be easily increased by using multimodal neuroimaging data and establish a solid and reliable lower bound for cognitive prediction in different domains.

## Materials and methods

### Participants

We used publicly available data in the S1200 release from the Human Connectome Project (HCP) Young Adult study (https://www.humanconnectome.org/study/hcp-young-adult). Out of the 1200 participants released, 1050 subjects had viable T1-weighted, resting-state fMRI, and diffusion MRI data. In addition, 22 subjects were discarded due to the presence of missing information in some of the response and confounder variables used in this study. The final dataset then comprised 1028 individuals (550 female, age range 22-37, mean $\pm \sigma_{age}$ = 22.73 $\pm$ 3.68 years).

### Predictor variables

Preprocessing steps included spatial artifact/distortion removal, surface generation, cross-modal registration and alignment to standard space and the automatic ICA-FIX denoising of functional acquisitions, among others (more details on these and other additional preprocessing steps can be found in [52]).

*Structural* predictors were composed of cortical thickness (CT) and surface area (CS) values of 360 regions in a multi-modal parcellation [53], and 65 features containing global, subcortical and other volume (VL) information, directly extracted from the *aseg.stats* freesurfer file of each subject. The estimated intracranial volume, which was also part of the *aseg.stats* file, was not considered a predictor but a confounder to adjust for (see below for more details).

*Functional* predictors were estimated from the resting-state data by first computing the averaged time series from the voxels within each of the 718 regions in a parcellation which

extends the aforementioned multi-modal atlas to include 358 subcortical regions [54]. Furthermore, this parcellation identified 12 different intrinsic functional networks, which included the well-known primary visual, secondary visual, auditory, somatomotor, cingulo-opercular, default-mode, dorsal attention and frontoparietal cognitive control networks; novel networks like the posterior multimodal, ventral multimodal, and orbito-affective networks; and the identification of a language network [54]. S6 Fig shows the spatial distribution of these 718 regions in the brain and their resting-state network assignment. Next, a functional connectome for each subject was built by calculating the z-transformed Pearson correlation coefficient between pairs of time series. Finally, the upper triangular elements were extracted to form the final vector of 257403 functional connectivity (FC) features per subject.

*Diffusion* predictors were represented by the local connectome fingerprint (LC), a structural metric that quantifies the degree of connectivity between adjacent voxels within a white matter fascicle defined by the density of diffusing spins [55]. The local connectome was computed by reconstructing the spin distribution functions (SDFs) in all white matter voxels in a common template space, previously derived from 842 subjects of the HCP dataset [28], using q-space diffeomorphic reconstruction [56] and sampling the quantitative anisotropy [57] at peak directions within each voxel. This produces a fingerprint vector of 128894 fibers across the entire brain for each subject. These features were obtained using DSI Studio (http://dsi-studio.labsolver.org), an open-source toolbox for connectome analysis from diffusion imaging.

## Response variables

A subset of seven cognitive test scores available in the HCP repository were used as response variables [58]. Each of these measures assesses the individual performance in cognitive domains that are different to a greater or lesser extent. In particular, we selected: (a) the Unadjusted scale NIH Toolbox Cognition Total Composite Score, which provides a measure of *global cognitive function*, (b) The Unadjusted NIH Toolbox Cognition Fluid Composite Score for *fluid intelligence*, (c) The Unadjusted NIH Toolbox Cognition Crystallized Composite Score for *crystallized intelligence*, (d) the area-under-the-curve (AUC) for Discounting of $200 in a Delay Discounting Test for *impulsivity*, (e) the total number of correct responses in a Variable Short Penn Line Orientation Test for *spatial orientation* assessment, (f) the Total Number of Correct Responses in a Penn Word Memory Test, which aims at testing *verbal episodic memory*, and (g) sensitivity in a Short Penn Continuous Performance Test for *sustained attention* performance. The main descriptive statistics for these variables can be found in Table 1 and their sample distributions and similarity between scores in S7 Fig.

## Prediction models

The prediction of each cognitive score was carried out adopting a transmodal approach to stacking learning. Stacking belongs to the ensemble paradigm in machine learning and it is based on a multi-level training in which predictions from a given set of models are combined to form a new meta feature matrix [22, 24]. This new matrix can be then fed into a new model for final predictions or passed to a successive and intermediate learning level.

Our transmodal scenario consisted of two-levels of learning and differs from usual stacking approaches in that each predictive model comes from training separately our different groups of features (called channels), each corresponding to the resting-state connectome, cortical thickness attributes, cortical surface areas, global and subcortical volumetric information and the local connectome fingerprints.

The entire prediction modeling procedure is illustrated in Fig 1. First, we split the data into training and test set (see next section for more details on the cross-validation procedure for

model performance assessment). We then adjusted for the effect of the intracranial volume on the response variable computing the residuals from a linear regression model estimated using only the training set information in order to avoid any data leakage. Next, a 5-fold cross-validation was applied to the observations in the training set using a principal component regression model with a L1 regularization term (LASSO-PCR) on each channel. This estimator constitutes a pipeline with the following sequential steps:

1. Dimensionality reduction by PCA to the input matrix of features $X$ of each channel:

$$X = USV^T \, , \tag{1}$$

   where the product matrix $Z = US$ represents the projected values of $X$ into the principal component space and $V^T$ an orthogonal matrix whose row vectors are the principal axes in feature space.

2. Regression of the response variable $y$ onto $Z$, where the estimation of the $\beta$ coefficients is subject to a L1 penalty term $\lambda$ in the objective function:

$$\hat{\beta} = arg \ min_\beta \{ \|y - Z\beta\|^2 + \lambda\|\beta\| \} \tag{2}$$

3. Projection of the fitted $\hat{\beta}$ coefficients back to the original feature space to produce a weight map $\hat{w} = V\hat{\beta}$ used to generate final predictions $\hat{y}$:

$$\hat{y} = X\hat{w} \tag{3}$$

This inner cross-validation loop (acting only on the training set) was used so as to determine the optimal value of $\lambda$ and simultaneously generate out-of-sample predictions that are to be used as the new training data at the subsequent learning level. After training each LASSO-PCR model with the optimal level of shrinkage, single-channel predictions from the test set were aggregated across all channels.

At the second level, both out-of-sample predictions from the training set and predictions on the test set were stacked across the five channels to form a new training and test set. A new LASSO regression model was then optimized and fitted to produce a final prediction. The motivation to use a LASSO model with this small number of features (the number of single-channels) at this learning stage was to perform a feature selection as well, that automatically selected and weighted how much each channel contributed to the best final prediction. Additionally, at this stage we imposed a non-negative constraint to the determination of the regression coefficients due to its effectiveness in increasing the performance by stacked regressions [59] and to aid the interpretation of the contributions from the single-channels. Finally, the performance of this prediction was estimated through the coefficient of determination $R^2$ and the mean absolute error (MAE). All this framework was implemented using scikit-learn [60].

## Cross-validation strategy for performance assessment

Training and test sets were obtained by splitting the data into 70% and 30% of the observations respectively. In order to estimate the predicted metrics taking into account different partition seeds, a Monte Carlo cross-validation was employed. In particular, we generated 100 random splits and obtained the model performance in each of these partitions. A final cross-validated performance was then reported using the median and its confidence intervals, computed by a percentile bootstrapping (1000 bootstrap samples) at a significance level $\alpha = 0.05$. The choice

of the median was particularly suitable when summarizing the contributions of each single-channel, since the stacked LASSO model produced sparse solutions.

To note, when generating the different training and test set partitions, we took into account the fact that the HCP database includes pairs of monozygotic and dizygotic twins, which can give rise to overoptimistic results if these pairs do not fall together in the same partition set (either the training or the test set). Our dataset included 419 observations with twin zygosity information as verified by genotyping. This odd number is due to some of the twins being discarded during the data preparation process (see previous sections). As a consequence, each set of twins were assigned with the same identification label and this information used to generate randomized train/test indices that guaranteed that they were always together in the same partition set. We accomplished this using the *model_selection.GroupShuffleSplit* class of scikit-learn.

## Confounding adjustment

We adopted a hybrid strategy for confounder adjustment in the predictions depending upon the variables whose effect we were controlling for, which we grouped into neuroimaging and mediating confounders respectively.

In the first group, we considered the total intracranial volume, extracted directly from the Freesurfer outputs for each subject, as the neuroimaging measure with a clear effect on the brain architecture properties and which is known to be closely related to cognitive performance. One possible way of controlling for this variable could be to include it as a covariate in our prediction models. However, this approach would imply mixing measures of different modalities in the feature matrix of each channel. As a consequence, we decided to partial out its effect from the response variables, for which we employed linear regression models whose intercept and slope coefficients were estimated using only the training sets in order to avoid any data leakage.

In the second group, we considered those variables that are not neuroimaging measures but can mediate an effect between these and cognitive performance, e.g., demographic factors. Since our stacking approach deals naturally with measures of different modalities, we thus decided to treat this group of variables as a different modality constituting an additional channel in our predictive framework. If the predictions from this confound channel were to carry most of the variability of a particular response variable, then our second level LASSO model should automatically shrink away the predictions from the rest of neuroimaging channels. That is because the L1 constraint chooses the best representative feature out of a set of correlated features. This strategy goes in accordance with recent work controlling for confounders on the level of machine learning predictions, allowing for estimating the portion of performance that can be explained by confounds and the portion of performance independent of confounds [61]. In our case, we considered sex, age and education level in this group.

## Weight maps of feature relevance

Our Monte Carlo cross-validation procedure allowed us to generate a distribution of LASSO weights that quantifies the relative importance of each channel to cognitive performance prediction. Thus, for each single-channel with a significant contribution to stacking at $\alpha = 0.05$, we fit a LASSO-PCR estimator to the **entire** set of observations and extracted their pattern of weights $\mathcal{W}$. It is intuitive to view these decoding weight maps as approximating their encoding representations, such that the weights could be deployed as models for predicting the same response variables on independent external datasets, with an expected accuracy provided by our reported cross-validation median performances. However, these maps emerge from models that attempt to extract neural information from data (also known as *backward* or *decoding*

models) and therefore, their interpretation regarding the importance of the brain properties and cognition could be problematic [16]. This is due to the fact that that large weights in the prediction models may appear as a consequence of the role of certain features simply to suppress noise in the data, although no direct relation between these features and the response variable exists *per se*. Consequently, following the recommendations given in [16], we interpreted the link between cognitive performance and the different brain properties by transforming these feature maps to *encoding patterns* $\mathcal{A}$ as follows

$$\mathcal{A} = \Sigma_X \mathcal{W} \Sigma_Y^{-1} \, , \tag{4}$$

where $\Sigma_X$ is the sample covariance matrix between the input features in each single-channel and $\Sigma_Y$ the sample covariance matrix between the response variables. Since we are dealing with each cognitive response variable separately, this latter matrix would simply correspond to a scalar constant representing the variance. As a consequence, we omitted such a quantity from the encoding patterns computation because it does not change the interpretation of each feature's relative importance with cognition.

## Stacking bonus

In order to formally compare the integrated model, *i.e.*, the model that integrates predictions across modality, against the single-channel predictions, we defined a stacking bonus score $\mathcal{B}$ which reads

$$\mathcal{B} \equiv R^2_{stacking} - <R^2>_{single} \, , \tag{5}$$

where $R^2_{stacking}$ is the out-of-sample coefficient of determination from the second-level LASSO learning to the stacked predictions and $<R^2>_{single} = \frac{1}{5}\left(R^2_{FC} + R^2_{CS} + R^2_{CT} + R^2_{VL} + R^2_{LC}\right)$ is the average performance across individual modalities. Therefore, this quantity aims to estimate the difference in performance between the joint model and the average across its parts, which in our case correspond to the different brain measurement channels.

Since the above definition may lead to very optimistic bonuses in the presence of poor modalities, we finally adopted a more conservative definition by expressing this difference just against the best single-channel performance $R^2_{best}$ as follows

$$\mathcal{B} \equiv R^2_{stacking} - R^2_{best} \tag{6}$$

## Supporting information

**S1 Fig. Single-channel and stacked performances to predict cognition.** Mean absolute errors (MAE) between the observed and predicted values of seven cognitive scores using each brain measurements separately and together by stacking their predictions. In red the scenario that yields the maximum score.
(TIF)

**S2 Fig. Resting-state connectivity weight patterns.** Encoding weight maps from the resting-state connectivity matrix coefficients for the two cognitive areas (global cognitive function and fluid intelligence) in which these attributes significantly contributed to stacking. Red and blue colors display positive and negative weights respectively. Loadings have been scaled to a [-1, 1] range without breaking the sparsity using the function *maxabs_scale* in scikit-learn.
(TIF)

**S3 Fig. Prediction pattern similarity between cognitive domains.** For each brain measurement, the Pearson correlation coefficient is computed to assess the similarity between the brain correlates of each cognitive score in which stacking led to a significant performance enhancement. A cross along domains for a given brain measurement indicates that the LASSO-PCR model failed to keep any feature during the optimization process.
(TIF)

**S4 Fig. Contributions of each single-channel in the stacked model in the presence of confounders.** Across the 100 different data splits, the weight distribution assigned to the out-of-sample predictions of each brain measurement by the stacked LASSO model that includes also a channel for confounders (gender, age and education level) in those original cognitive scores in which stacking significantly improved the overall performance.
(TIF)

**S5 Fig. Distributions for new scores of sustained attention and impulsivity.** Univariate distributions for the Five Factor Model (NEO-FFI) score for extraversion, a proxy for impulsivity, and the Short Penn CPT median Response Time For True Positive Responses for sustained attention.
(TIF)

**S6 Fig. A multi-modal brain parcellation.** A parcellation consisting of 718 regions, for which 360 are cortical regions and the remaining subcortical. In colors, their assignment to 12 major resting-state networks as provided in [54].
(TIF)

**S7 Fig. Pairplot between cognitive scores.** A pairplot where the diagonal shows the univariate distributions for the response variables used in our study and the off-diagonals the similarity between them, visualized by means of scatterplots and quantified using Pearson correlation coefficients.
(TIFF)

**S1 Table. Average positive weights of local connectome features per major tract.** After rescaling the weights of local connectome features to a range [-1, 1] without breaking their sparsity, average positive loadings within each major tract in a population-based atlas [28] were computed. An entry with a line mark denotes the lack of positive weights within that tract. L≡Left hemisphere, R≡Right hemisphere.
(XLSX)

**S2 Table. Average negative weights of local connectome features per major tract.** After rescaling the weights of local connectome features to a range [-1, 1] without breaking their sparsity, average negative loadings within each major tract in a population-based atlas [28] were computed. An entry with a line mark denotes the lack of negative weights within that tract. L≡Left hemisphere, R≡Right hemisphere.
(XLSX)

**S3 Table. Weights of volumetric properties.** Loadings of global and subcortical volume features for predicting the score of a Delay Discounting test, which assesses impulsivity abilities. The names of the features are the same that can be found in the *aseg.stats* file from Freesurfer. Weights have been scaled to a [-1, 1] range without breaking the sparsity using the function *maxabs_scale* in scikit-learn. Four features (Left-WM-hypointensities, Right-WM-hypointensities, Left-non-WM-hypointensities, Right-non-WM-hypointensities) are not shown here

since they had zero variance and were therefore discarded by the estimator during the fitting process.
(XLSX)

## Acknowledgments

Data were provided by the Human Connectome Project, WU-Minn Consortium (Principal Investigators: David Van Essen and Kamil Ugurbil; 1U54MH091657) funded by the 16 NIH Institutes and Centers that support the NIH Blueprint for Neuroscience Research; and by the McDonnell Center for Systems Neuroscience at Washington University.

## Author Contributions

**Conceptualization:** Javier Rasero, Amy Isabella Sentis, Fang-Cheng Yeh, Timothy Verstynen.

**Data curation:** Javier Rasero, Fang-Cheng Yeh.

**Formal analysis:** Javier Rasero.

**Investigation:** Javier Rasero, Amy Isabella Sentis.

**Methodology:** Javier Rasero, Timothy Verstynen.

**Project administration:** Timothy Verstynen.

**Software:** Javier Rasero.

**Supervision:** Fang-Cheng Yeh, Timothy Verstynen.

**Visualization:** Javier Rasero.

**Writing – original draft:** Javier Rasero, Amy Isabella Sentis, Fang-Cheng Yeh, Timothy Verstynen.

**Writing – review & editing:** Javier Rasero, Amy Isabella Sentis, Fang-Cheng Yeh, Timothy Verstynen.

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
