## [Decision Letter · Decision Letter 0]

20 Oct 2020

Dear Dr Rasero,

Thank you very much for submitting your manuscript "Integrating across neuroimaging modalities boosts prediction accuracy of cognitive ability" for consideration at PLOS Computational Biology.

As with all papers reviewed by the journal, your manuscript was reviewed by members of the editorial board and by several independent reviewers. In light of the reviews (below this email), we would like to invite the resubmission of a major revision that takes into account the reviewers' comments and requests for additional analyses.

We cannot make any decision about publication until we have seen the revised manuscript and your response to the reviewers' comments. Your revised manuscript is also likely to be sent to reviewers for further evaluation.

Sincerely,

Emma Claire Robinson

Associate Editor

PLOS Computational Biology

Samuel Gershman

Deputy Editor

PLOS Computational Biology

Reviewer's Responses to Questions

**Comments to the Authors:**

Reviewer #1: In this paper, the authors investigate prediction performance for behavioral measures in a large open dataset of young healthy adults when using several neuroimaging modalities with a transmodal learning approach. Overall the topic is of interest for the community and the paper is well-written, the main conceptual ideas are well-communicated for a large readership. I nevertheless have methodological concerns that should be addressed in order for the paper to be further considered for publication.

Main first main concern is related to the interpretation of the features contributing to the prediction based on the weight in the LASSO. This type of postdoc interpretation has been shown to be dangerously misleading. Clear demonstration, discussion and recommendations are provided in Haufe et al. (2014).

My second main comment pertains to confounding factors. It seems that the authors haven’t carefully controlled for other factors than age, in particular they did not control for the effect of brain size (with TIV or ICV) which often spuriously influence covariance between neuroimaging features and behavioral features (partly through the effect of gender).

My third main comment is related to the selection of the behavioral scores of interest. The choice was not supported by any arguments. It is good that the authors tried to cover different behavioral domains, however, many of the scores appears actually unpredictable based on brain data. The distribution of the unpredictable behavioral measures could be questioned. The authors gave some specific summary measures but could they provide visual illustrations of distribution (basically histograms) while taking care of showing the whole range of possible values in each test on the x axis? I suspect that some scores have ceiling effect and lack relevant interindividual variability that would be needed to get any predictive power as partly discussed by the authors on line 422. Some scores could be replaced by scores for which relatively decent prediction performance have been reported in other studies. For example, impulsivity could be replaced by extraversion.

Furthermore, it was not clear whether the family structure was taken into consideration in the cross-validation scheme? having family members split (in particular twins) across folds could result in overoptimistic performance (the model trains on a group of subjects would work better on the relatives of those subjects than in unrelated individuals).

Minor comments:

Introduction:

Introduction should be more balanced: machine learning approaches suffers from a lack of interpretability and despite all the promises and perspectives that have been suggested by several review pieces, the contribution of these approaches to our understanding of brain-behavior relationships remains currently very limited, if not null.

Issues related to false positive, replicability, and sample size are illustrated and discussed along the same lines in Kharabian Masouleh, Eickhoff, Hoffstaedter, Genon, and Alzheimer's Disease Neuroimaging (2019).

What is the meaning of “A multi-modal cognitive phenotype”? A cognitive phenotype refers to the expression of a specific cognitive pattern, for example, low episodic memory performance together with low spatial navigation performance and low extraversion score in a group of individuals. Here, the authors are actually referring to a multi-modal brain phenotype that related to cognition.

Results:

Line 158: “with the former contributing significantly more than the latter”, what is the statistical test here ?

“Such diverse patterns of contributions from single-channels to the stacked model may be partially affected by the L1 regularization term dealing with the shared variance between predictions” this actually shows that the interpretation of the weights following LASSO should be avoid.

Line 179, “anterior parietal areas” is relatively fuzzy, to which anatomical regions do the authors actually refer to?

Overall, cortical thickness and cortical surface weights go against what could be expected from the neuroscience literature: positive associations with high-level cognitive scores are found in regions around the central sulcus and even in visual cortex for fluid intelligence, while negative associations are found in most of the association areas. A similar concern can be raised for association of global cognitive ability with brainstem pathway. How did the authors explain these puzzling findings ?

Discussion:

Line 378: “mean global BOLD signal, which is supposed to strengthen…” this sentence is confusing, currently it can read as global signal strengthen the association while this is the global signal regression procedure that strengthens the predictive power.

Reference:

Haufe, S., Meinecke, F., Görgen, K., Dähne, S., Haynes, J.-D., Blankertz, B., & Bießmann, F. (2014). On the interpretation of weight vectors of linear models in multivariate neuroimaging. Neuroimage, 87, 96-110.

Kharabian Masouleh, S., Eickhoff, S. B., Hoffstaedter, F., Genon, S., & Alzheimer's Disease Neuroimaging, I. (2019). Empirical examination of the replicability of associations between brain structure and psychological variables. Elife, 8. doi:10.7554/eLife.43464

Reviewer #2: Integrating across neuroimaging modalities boosts prediction accuracy of cognitive ability

Javier Rasero, Amy Isabella Sentis, Fang-Cheng Yeh, Timothy Verstynen

Summary

This study tests whether the prediction accuracy of cognition can be improved by combining multiple imaging modalities. Using data from the human connectome project, the study combines features from diffusion, resting state, and structural MRI in a stacked model. Separate Lasso regularized regression models are first fit for each modality, and then the predictions from all modalities are used as input features into a second level lasso regression. The results show improvements of the combined model over and above each individual modality prediction.

General

This paper tackles a worthwhile question in a straightforward way. It is nice to see the authors avoiding overly complex models, and the article is written very clearly and reproducibly. I only have a few fairly minor comments. Please find my detailed comments below, approximately in order of importance.

1. The individual modality models are first optimized in cross-validation loops, and then the cross-validation of the second level model is performed separately. I wonder if it would be better to optimize both the first regression models and the second level model within the same cross-validation loop. Analogous to figure 1 in Dinga et al (https://doi.org/10.1016/j.nicl.2019.101796), we could think of the modality-specific models as the feature selection for the second level model. I would be interested in the authors’ thoughts on all-in-one cross-validation versus the separate cross-validation loops that are currently in the manuscript.

2. The figures of the resting state weights in figure 4 are not particularly informative. It seems that each of the networks has similarly strong negative and positive contributions. Without knowing the spatial distribution of these, it is hard to interpret these results. Perhaps something like figure 2 in Smith et al (https://doi.org/10.1038/nn.4125) would be more informative?

3. In the participants section, I suspect that some of the details about the human connectome project are incorrect. There are two HCP’s, and I think the PIs described here are not for the HCP data that was used here. For the young adult HCP (https://www.humanconnectome.org/), the PIs are David Van Essen and Kamil Ugurbil. Also, the universities that participated in the collaboration reflect the wrong HCP project.

Reviewer #3: This paper provides evidence that combining different MRI modalities accounts for a slightly larger portion of variance in cognitive abilities and attributes than each of modality alone. To test this idea, the authors analyzed a large dataset from the Human Connectome Project, which includes multi-modal MRI measurements and measurements of a battery of cognitive and behavioral assessments. The results demonstrate the combining features from different modalities through a stacked twice-regularized linear regression model can provide a small (1-4%) but consistent boost in variance explained, compared to the best uni-modal regularized regression model. These results are important because, while projects such as HCP are collecting multi-modal data, it is still rather unusual that data are combined across modalities to account for behavioral variance. This result motivates further exploration of these relationships, and shows the way towards a principled approach to do this, with structured/interpretable machine learning approaches.

Overall, the paper is well written, the analysis methods are of high quality and the interpretation of the results is sound. The high level of technical proficiency and transparency in sharing of code and data are particularly remarkable, and provide clarity about the methods and results well beyond the current standards of the field. The use of non-parametric permutation statistics for analysis of statistical significance is also a strength. That said, I think that the some of the choices made in setting up the comparisons are not clear or not sufficiently well-motivated. Testing and comparing alternative approaches to this analysis may provide more direct and possibly also more powerful evidence of the original hypothesis.

My main concern is that, beyond reference to previous work that used such an approach (Liem et al., 2017 and the original Wolpert work), it is not clear why the stacking architecture is chosen here, and whether this choice is beneficial. In particular, among the examples cited, this is the first to use linear models with the stacking architecture, and it is not clear that this combination makes the most of the integration across modalities. For example, one hypothesis about the relationship between brain features and complex cognitive constructs is that many small and distributed biological effects all add to explain the variance in cognition (see e.g., https://www.biorxiv.org/content/10.1101/2020.09.01.276451v1 and references therein). It seems that the stacking architecture is designed to specifically suppress these kinds of cumulative effects, particularly across modalities.

An alternative to the stacking model would have been to compare the modality-specific models to a LASSO-regularized model that includes all of the features across all modalities. It is possible that this would have (1) provided even higher variance explained and (2) provided more information about redundancy and complementarity between the models. For example, some of the data from different modalities pertains to similar anatomical locations (e.g., cortical thickness and resting-state connectivity of the same parts of cortex). The current model architecture doesn't allow the model to aggregate these across modalities, and improve the estimation SNR through this aggregation. Meanwhile, a LASSO over all of the features would help adjudicate whether different features extracted from the same anatomical location are redundant, and whether one is reliably selected over the other, or whether they combine to provide higher accuracy. This is because the stacked architecture strictly limits the interactions that can take place between the features at the first level, only allowing features that are important at the unimodal level to come through.

This might also adversely affect the results presented: the stacked model, when using LASSO regularization, should produce winner-take-all kinds of behavior and indeed, in examining Figure 3, it looks as though weights on the different modality channels vary by quite a bit, even between samples that differ only by approximately 2% of the data (20 subjects / ~1000 subjects). This is worrisome, because it suggests that even with the large amount of data that was analyzed here, the stacked model is highly variable in its weighting of the different modalities, and thus that small shifts in the noise can drive weights extremely from one modality channel to another modality channel. This may be the reason for bimodal weight distributions in Figure 3. Granted, the data doesn't explicitly show this, so this is an inference, but one that could be refuted or validated.

A question that arises is then: even if you stick to this stacked model, why is LASSO regularization even needed for the second level of analysis? It seems neccessary for the first level models, where the number of features is very large, sometimes even much larger than the number of subjects. But the second level model has only 5 features. It seems like the authors could fit an unregularized linear model to these features, which may prove more stable, and possibly also more accurate: essentially an optimally-weighted average of the predictions from the lower level. One kind of information that could help understand whether LASSO is required at this stage is a bit more insight into the optimal regularization parameter that was found at that level.

Relatedly, I found the description of the cross-validation procedure in the Methods to be a bit confusing. After looking through the code provided, I am convinced that this is done correctly, but the description in L555-L562 could raise doubts. In particular, I think that it would be good to clarify that the 70% mentioned on L557 are selected from within the 70% that were previously (L531) designated as the training set. If I understand correctly from my examination of the code, this is a form of nested cross-validation done only within the training set (within cell 49 of https://github.com/CoAxLab/multimodal-predict-cognition/blob/master/notebooks/05-predictions.ipynb). But the text around that part of the Methods is a bit ambiguous, and a reader might wonder whether the 70% in L557 are from a new partition of the entire dataset, which could risk leakage from the test set.

Furthermore, one more suggestion. If the hierarchical nature of the stacked model is what the authors consider to be important in assessing the contributions of different modalities (i.e., the weights presented in Figure 3), there are still models that would provide this information, while still allowing the features at the first layer to interact with each other. For example, the authors might consider using a multi-layered perceptron, with L1 regularization (at the first, or both levels), to fit a hierarchical model. Again, one might expect better model performance, and more stability of second-level parameters across folds, with additional interpretability of relationships between the weights. I'll admit that this might be far out of scope for the current work, and could be discussed as potential future work. But at the very least, I think that it would improve the paper to add a comparison of the stacked architecture to simpler alternative model architectures (at least just a single-level PCA-LASSO or LASSO), which could demonstrate the utility of the stacking architecture in this setting.

Finally, a question regarding adjustments to the original variables. The authors chose to adjust the cognitive scores for age. I wonder whether similar adjustments would be applied to the brain data? In particular, whether cortical regional surface area should be normalized to overall cortical surface area.

Minor comments:

In the Introduction, the authors cite the WU-Minn HCP, but the Methods point to the LONI DB and to the USC/MGH HCP. Based on the statistics of the data (e.g., number of participants), I do think that the WU-Minn HCP was used. Is that the case here?

L57: Though this sentence points to the limitations of a lot of unimodal work, it's not always an implicit assumption that analyzing one type of data obviates other data. In other words, this is a bit strongly stated here.

L353: "begs the question" does not mean that the question remains unanswered. Consider changing into "...this raises the question... " or some-such.

The sentence that starts on L382 is a bit unfortunately worded, because it's unclear which of the two (Pearson's or COD) is being referred to in the end of the sentence (Pearson's, presumably).

I think that the authors used equation 5 to calculate the stacking bonus. So, it is not clear what we are supposed to make of equation 4. I also do not really understand how this relates to information and synergy among information transmitting channels. The text says "...joint system may convey more information than just the sum of its parts". I have two problems with this statement: (1) I don't think that R^2 is a good measure of information and (2) that's the average, not the sum. In other words: the connection to information theory is tenuous.

In looking at the GitHub repo, I noticed a commit "remove restrictive info". If this is information that should not be publicly available, the authors might want to rebase commits that include this file out of the history of their repository (e.g., https://github.com/CoAxLab/multimodal-predict-cognition/commit/2dab4b481c62314a09850de13ca92042a9cfa598)

What are the units on the scales of the X axes in Figure S1? Are these axes somehow comparable to each other?

**Have all data underlying the figures and results presented in the manuscript been provided?**

Reviewer #1: None

Reviewer #2: Yes

Reviewer #3: Yes

PLOS authors have the option to publish the peer review history of their article (what does this mean?). If published, this will include your full peer review and any attached files.

Reviewer #1: No

Reviewer #2: **Yes: **Janine Bijsterbosch

Reviewer #3: No
---

## [Decision Letter · Decision Letter 1]

10 Feb 2021

Dear Dr Rasero,

We are pleased to inform you that your manuscript 'Integrating across neuroimaging modalities boosts prediction accuracy of cognitive ability' has been provisionally accepted for publication in PLOS Computational Biology.

We also strongly recommend that you adjust the publication to address the minor points made by the reviewers which would likely strengthen the publication.

Best regards,

Emma Claire Robinson

Associate Editor

PLOS Computational Biology

Samuel Gershman

Deputy Editor

PLOS Computational Biology

Reviewer's Responses to Questions

**Comments to the Authors:**

Reviewer #1: The authors have taken into account all my suggestions and significantly strengthened their manuscript. I noted nevertheless that regressing out confounds has importantly reduced the prediction performance of their model, which could now appear relatively limited for naïve readers not aware of the challenges of predicting interindividual differences in behaviour using neuroimaging data. I would therefore suggest that the authors briefly highlight and discuss this point. They could for example have the previous version of Figure 2 and the current one side by side.

Some additional suggestions:

“Aspects of the brain” should be replaced by “neurobiological properties” (line 67)

Line 101: I am not sure “utility” is the right term here

Line 199: “crystallized intelligence (i.e., the ability to recall and use prior information)” is not the best definition of crystallized intelligence and can be misleading for the naïve reader as it can be assimilated to memory retrieval while crystallized intelligence refers more to a set of general knowledge (including vocabulary) acquired across experience and hence strongly related to education.

Reviewer #2: The revised manuscript has greatly improved and the authors have been thorough in their responses. All of my comments have been appropriately addressed.

Reviewer #3: I think that the article has substantially improved through this revision and I have no additional concerns.

I have a couple of really small comments, which the authors may choose to take or leave as they see fit:

1. On L654, I think that "...an hybrid..." should be "...a hybrid..."

2. In their acknowledgement to the HCP, the authors left the optional parenthetical "Data were provided [in part] by the Human Connectome Project". I think that all of the data shown here was from HCP, so that parenthetical can probably go away: "Data were provided by the Human Connectome Project".

**Have all data underlying the figures and results presented in the manuscript been provided?**

Reviewer #1: None

Reviewer #2: Yes

Reviewer #3: None

PLOS authors have the option to publish the peer review history of their article (what does this mean?). If published, this will include your full peer review and any attached files.

Reviewer #1: No

Reviewer #2: No

Reviewer #3: No

---

## [Editor Report · Acceptance letter]

27 Feb 2021

PCOMPBIOL-D-20-01715R1 

Integrating across neuroimaging modalities boosts prediction accuracy of cognitive ability

Dear Dr Rasero,

I am pleased to inform you that your manuscript has been formally accepted for publication in PLOS Computational Biology. Your manuscript is now with our production department and you will be notified of the publication date in due course.

With kind regards,

Alice Ellingham
